# Single-Cell Transcriptome of Wet AMD Patient-Derived Endothelial Cells in Angiogenic Sprouting

**DOI:** 10.3390/ijms232012549

**Published:** 2022-10-19

**Authors:** Natalie Jia Ying Yeo, Vanessa Wazny, Nhi Le Uyen Nguyen, Chun-Yi Ng, Kan Xing Wu, Qiao Fan, Chui Ming Gemmy Cheung, Christine Cheung

**Affiliations:** 1Lee Kong Chian School of Medicine, Nanyang Technological University, Singapore 636921, Singapore; 2Duke-NUS Medical School, National University of Singapore, Singapore 169857, Singapore; 3Ophthalmology & Visual Sciences Academic Clinical Program (Eye ACP), Duke-NUS Medical School, Singapore 169857, Singapore; 4Singapore Eye Research Institute, Singapore 169856, Singapore; 5Institute of Molecular and Cell Biology, Agency for Science, Technology and Research, Singapore 138673, Singapore

**Keywords:** age-related macular degeneration, blood outgrowth endothelial cells, sprouting angiogenesis, single-cell transcriptome, endothelial cell states

## Abstract

Age-related macular degeneration (AMD) is a global leading cause of visual impairment in older populations. ‘Wet’ AMD, the most common subtype of this disease, occurs when pathological angiogenesis infiltrates the subretinal space (choroidal neovascularization), causing hemorrhage and retinal damage. Gold standard anti-vascular endothelial growth factor (VEGF) treatment is an effective therapy, but the long-term prevention of visual decline has not been as successful. This warrants the need to elucidate potential VEGF-independent pathways. We generated blood out-growth endothelial cells (BOECs) from wet AMD and normal control subjects, then induced angiogenic sprouting of BOECs using a fibrin gel bead assay. To deconvolute endothelial heterogeneity, we performed single-cell transcriptomic analysis on the sprouting BOECs, revealing a spectrum of cell states. Our wet AMD BOECs share common pathways with choroidal neovascularization such as extracellular matrix remodeling that promoted proangiogenic phenotype, and our ‘activated’ BOEC subpopulation demonstrated proinflammatory hallmarks, resembling the tip-like cells in vivo. We uncovered new molecular insights that pathological angiogenesis in wet AMD BOECs could also be driven by interleukin signaling and amino acid metabolism. A web-based visualization of the sprouting BOEC single-cell transcriptome has been created to facilitate further discovery research.

## 1. Introduction

Age-related macular degeneration (AMD) is a major cause of irreversible blindness in older populations [1,2]. An exudative subtype of this disease, known as ‘wet’ AMD, is responsible for approximately 90% of AMD cases with severe vision loss [3]. Wet AMD is characterized by the anomalous vascularization of choroidal vessels into the subretinal space (choroidal neovascularization) with accompanying exudate accumulation and/or hemorrhage leading to vision loss [4,5,6]. During disease development, degenerative changes occur in the retinal pigmented epithelium, choroidal vessels, Bruch’s membrane, and photoreceptor layers that ultimately create a proinflammatory and angiogenic milieu favoring choroidal neovascularization [7]. Anti-vascular endothelial growth factor (VEGF) treatment is the gold standard therapy for wet AMD [8,9]. However, anti-VEGF treatment is effective in a subset of AMD patients and seems ineffective in the long-term prevention of visual decline [8,10,11,12]. Such findings warrant a need to illuminate its deeper molecular mechanisms, as well as VEGF-independent pathways.

In vivo models have been crucial to shed light on the molecular pathogenesis of wet AMD. A common animal model for wet AMD is the laser-induced choroidal neovascularization model, in which neovascularization of choroidal vessels in the retina is induced by laser exposure to the Bruch’s membrane [10,13]. While these murine models have provided valuable insights into mechanisms and therapeutic testing [13,14,15], the mouse eye is lacking a defined macula compared to humans [10,16,17]. Laser-induced choroidal neovascularization models may not fully capture the genetic and pathological complexities of human wet AMD [13,18]. On the other hand, numerous in vitro human-relevant models of wet AMD have been created from retinal pigmented epithelium [19,20,21]. Despite the involvement of choroidal vascular endothelium in the pathogenesis of both typical wet AMD and other clinical variants [7,22,23,24,25], few patient-derived vascular cell models of wet AMD have been developed for disease interrogation [26,27,28,29,30]. There remain knowledge gaps in the investigation of endothelial-specific mechanisms in wet AMD.

To elucidate the disease molecular signatures of human wet AMD endothelium during angiogenesis, we leveraged on our model of blood outgrowth endothelial cells (BOECs) developed from wet AMD patients. We and others have shown that BOECs can be derived from peripheral blood of patients and established as endothelial models recapitulating the morphological and functional characteristics of mature endothelial cells [31,32,33,34,35]. Here, we studied angiogenic behaviors of patient BOECs using a three-dimensional sprouting assay. As sprouting angiogenesis induces multiple endothelial phenotypes [36], we performed analysis on the single-cell RNA-sequencing (scRNA-seq) of sprouting BOECs to resolve the endothelial subpopulations. To facilitate data mining of wet AMD endothelial molecular hallmarks, we created a web-based visualization of sprouting BOEC single-cell transcriptome (https://christinecheunglab.shinyapps.io/human_wet_AMD_sprouting/, accessed and last updated on 10 October 2022).

## 2. Results

We have developed our BOEC models from peripheral blood mononuclear cell fractions isolated from wet AMD patients and normal controls [31]. Demographics of the donors from whom BOECs were used here have been included in Appendix A. These BOECs were subjected to a three-dimensional fibrin gel bead sprouting assay for 24 hours, displaying multiple sprouts on the beads with filopodia (Figure 1a), which were characteristic of endothelial cells responding to angiogenic cues [37]. To deconvolute the heterogeneity of angiogenic endothelial cells from patients, we dissociated sprouting BOECs in fibrin gel using nattokinase and prepared individual scRNA-seq libraries for sequencing (Figure 1a).

### 2.1. Human Blood Outgrowth Endothelial Cells Undertake Distinct Cell States during Sprouting Angiogenesis

To determine an optimal number of endothelial cell states, we took reference of the cellular phenotypes that could arise during angiogenesis. In vascular angiogenic sprouts, at least three types of cell specializations are present—tip cells that are the leading cells of the vascular sprout typically showing filopodia extensions, stalk cells that trail after tip cells and display active proliferation, and endothelial cells of the main vascular plexus [37]. We generated a clustering tree for 4 resolutions ranging from 0.04 to 0.1 using Seurat’s *FindClusters* function (Figure 1b, right panel). Based on the clustering hierarchy, Clusters ‘0′ and ‘1′ were transcriptomically distinct and represented the primary populations. At higher resolutions (0.08 and 0.1), Cluster ‘1′ remained independent from the other clusters, while Cluster ‘0′ diverged into Cluster ‘3′ and subsequently gave rise to Cluster ‘2′. Both Clusters ‘2′ and ‘3′ remained stable as the evolved populations. We were cognizant that tip and stalk cell specializations are transient and interchangeable, depending intricately on molecular cues (local VEGF, Notch signaling) [38,39,40]. Four transcriptomic clusters (Figure 1b, left panel) might reflect the dynamism of angiogenic phenotypes and potentially recapitulate a continuum of transitory cell states in the endothelial sprouts.

Upon derivation of cluster marker genes and functional enrichment analysis of these cluster markers (Figure 1c–f), we assigned the four endothelial cell states, Clusters ‘0′, ‘1′, ‘2′ and ‘3′, as vascular remodeling, proliferative, energetically primed and activated, respectively. Vascular remodeling is a process by which blood vessels restructure and rearrange through cell migration, cell growth, and degradation or synthesis of the extracellular matrix [41]. Under the gene ontology term ‘response to hypoxia’ (Figure 1c), vascular remodeling BOECs expressed genes such as VEGF-induced *PTGS2* (cyclooxygenase-2), an early response gene that leads to downstream proangiogenic effects [42,43], *THBS1* (thrombospondin-1), a hypoxia-induced modulator of vascular remodeling in pulmonary arterial hypertension [44,45], and *EDN1* (endothelin-1), a potent growth factor that induces the migration, proliferation and invasion of endothelial cells during angiogenesis [46]. The other enriched processes of ‘cell-substrate adhesion’ and ‘extracellular matrix organization’ (Figure 1c) emphasized matrix reorganization during cell migration. Substantiating the vascular remodeling phenotype of BOECs in this cluster, molecular functions such as ‘extracellular matrix structural constituent’ (consisting of collagen genes) and ‘integrin binding’ (Appendix A) suggested that vascular remodeling BOECs might be participating in the initiation of angiogenesis.

Another distinct primary population, the proliferative BOECs displayed many enriched biological processes relating to mitotic cell division and microtubule binding (Figure 1d and Appendix A). Among the top marker genes of proliferative BOECs (Figure 1g), *MKI67*, *CENPF*, and *TOP2A* are well-known proliferation markers and *ASPM* encodes a mitotic spindle protein [47,48]. It is plausible that these proliferative BOECs present in the fibrin gel sprouting assay were inherently in the dividing phase of the cell cycle or as part of vascular sprout formation [37].

Energetically primed BOECs had evolved from the vascular remodeling BOECs. The enriched processes ‘ATP metabolic process’ and ‘oxidative phosphorylation’ (Figure 1e) revealed that these energetically primed BOECs were switching out from a quiescent state to undergo the most energetically demanding phase of angiogenesis. The increased expression of cluster markers *ENO1* (enolase 1) and *TPI1* (triosephosphate isomerase 1) (Figure 1g), both glycolytic enzymes [49,50], suggested that glycolytic pathways were also prevalent in these cells. Glycolytic metabolism is increased in angiogenic endothelial cells to generate energy for either cell motility or proliferation [51]. In line with this, it has been reported that VEGF can stimulate both glycolysis and mitochondrial respiration in ECs in vitro [52]. Interestingly, energetically primed BOECs also demonstrated the biological process ‘negative regulation of G2/M transition of the cell cycle’ (Figure 1e), which was mainly represented by genes encoding for proteasomal subunits. The degradation of cyclins and other cell cycle regulators by the 26S proteosome is essential to the oscillation of cell cycle proteins and consequently regulation of the cell cycle [53,54]. Specifically, as indicated by the biological processes ‘SCF-dependent proteasomal ubiquitin-dependent protein catabolic process’ and ‘anaphase-promoting complex-dependent catabolic process’ (Figure 1e), these BOECs seemed to involve the ubiquitin ligases SCF complex, which degrades G1 Cdk inhibitors to control entry into S phase, and anaphase-promoting complex, which is essential for exit from mitosis [54]. Inactivation of the anaphase-promoting complex also enables its substrates the glycolytic enzyme PFKFB3 and glutaminase to upregulate glycolysis and the biosynthesis of macromolecules for cell cycle progression [55,56], therefore linking the upregulation of metabolic processes with cell cycle regulation observed in energetically primed BOECs.

Activated BOECs might have temporally developed from the vascular remodeling BOECs. Biological processes such as ‘cellular response to tumor necrosis factor’ and ‘cellular response to lipopolysaccharide’ (Figure 1f), as well as increased gene expressions of NFκB pathway mediators, were characteristic of activated endothelium [57]. Tumor necrosis factor signaling could be inducing the specialization of tip cell-like phenotypes in this population [58]. Furthermore, molecular function analysis revealed ‘cytokine receptor binding’ and ‘chemokine activity’, suggesting that activated BOECs were actively involved in binding of inflammatory molecules and stress response (Appendix A). After determining the overarching cell states, we proceeded to analyze wet AMD versus normal control endothelial cells.

### 2.2. Wet AMD Endothelial Cells Reveal Pathological Angiogenesis Hallmarks

Next, we performed differential expression analysis within each population to compare the transcriptomic profiles of wet AMD and normal sprouting BOECs. We found that across all populations, wet AMD BOECs consistently featured endothelial cell migration and regulation of angiogenesis, while normal BOECs demonstrated cell-substrate adhesion (Figure 2a–d). In the primary populations, extracellular matrix organization and/or degradation were upregulated pathways in wet AMD BOECs (Figure 2a,b), suggesting that matrix degradation mediated their enhanced migration and angiogenesis. Since early stages of angiogenesis require the degradation of extracellular matrix for migration and proliferation of vascular sprouts [59], wet AMD BOECs could adopt a more angiogenic and migrative phenotype than normal control cells. Normal BOECs, on the other hand, reflected cell-substrate adhesion as a consistent feature across the proliferative, energetically primed and activated populations (Figure 2b–d). The process of cell substrate adhesion seemed to be mediated by ‘integrin signaling pathways’ that included genes such as *ITGA3*, *ITGA4*, *ITGAV* and *ITGB5* (Figure 2c). Specific integrins are known to be upregulated in endothelial cells during VEGF/TGFβ-mediated angiogenesis [60,61]. Both α_v_β_3_ and α_v_β_5_ are among the key mediators of physiological angiogenesis. Ablation of β_3_ and β_5_ integrins actually resulted in increased pathological angiogenesis and tumor growth [62], indicating that normal BOECs possessed integrins that were unlikely the drivers of neovascularization under pathological states.

Focusing on the primary populations—vascular remodeling and proliferative BOECs, wet AMD BOECs had upregulated ‘interleukin-4 and interleukin-13 signaling’ (Figure 2a,b). In line with these findings, concentrations of interleukin-4 (IL-4) and interleukin-13 (IL-13) in serum and aqueous humor samples were significantly elevated in AMD patients compared to healthy controls [63,64,65]. The receptor of IL-4, IL-4Rα, stimulated pathological angiogenesis of bone marrow-derived endothelial progenitor cells, contributing to choroidal neovascularization formation [65]. Furthermore, both IL-4 and IL-13 induced mRNA expression of *VCAM1* in vascular endothelial cells which is involved in VCAM-1/integrin α_4_ adhesion during inflammation and angiogenesis [66]. Supporting this notion, *VCAM1* was one of the wet AMD-upregulated genes in the primary populations (Figure 2a,b). These data suggest that IL-4 and IL-13 signaling may be responsible for enhanced pathological angiogenesis, independent of VEGF, in wet AMD BOECs.

Normal BOECs in the proliferative population appeared to be driven by ‘signaling by VEGF’, ‘signaling by PDGF’ and ‘Eph-Ephrin signaling’ (Figure 2b). Firstly, among the upregulated pathway ‘signaling by VEGF’ in normal BOECs, *FLT1* is a decoy receptor binding with high affinity to VEGFA but acts to suppress VEGF signaling in the vascular endothelium [67]. During tip-stalk competition in angiogenic sprouting, cells that express lower FLT1 levels gain the advantage of becoming and maintaining the tip cell phenotype [39]. Correspondingly, the downregulated *FLT1* in wet AMD BOECs might favor excessive tip cell formation. Secondly, upregulated Eph/Ephrin signaling in normal BOECs is responsible for modulating angiogenesis and arterio-venous remodeling to form proper capillary networks [68,69]. Some ephrins, such as *ephrinA1*, are also negative regulators of proliferation [70]. Therefore, reduced ephrin signaling in wet AMD BOECs might exert a stimulatory effect on proliferation, contributing to their proangiogenic phenotype.

Moving onto the evolved populations—energetically primed and activated BOECs, wet AMD BOECs had upregulated pathways such as ‘response of *EIF2AK4* (*GCN2*) to amino acid deficiency’ and ‘metabolism of amino acids and derivatives’ (Figure 2c,d). *EIF2AK4* is a sensor of amino acid starvation that induces ATF4 signaling resulting in the secretion of angiogenic VEGF by vascular endothelial cells [71]. We postulated that wet AMD BOECs experienced amino acid starvation, thus activating *EIF2AK4* to increase VEGF production. Interestingly, amino acid deprivation-induced VEGF expression is evident in human tumors and tumor cell lines, suggesting that the EIF2AK4/ATF4 axis stimulates tumor angiogenesis. Wet AMD BOECs in energetically primed population also upregulated genes for amino acid metabolism (Figure 2c). We expected that energetically primed BOECs could gain additional biomass and energy during angiogenic sprouting. To increase biomass, proliferating/migrating endothelial cells increased glycolytic flux, fatty acid metabolism and amino acid metabolism compared to quiescent endothelial cells [72]. Furthermore, it has been reported that the switch from quiescence to angiogenic endothelial states in mouse choroidal neovascularization was accompanied by metabolic transcriptome plasticity [73]. Hence, wet AMD BOECs might have upregulated amino acid metabolism pathways to meet biomass demands in their enhanced propensity to undergo hypersprouting. We have distilled the molecular hallmarks comparing wet AMD and normal BOECs in Figure 2e.

### 2.3. Benchmarking Human Sprouting BOECs to In Vivo Choroidal Neovascularization Profiles

In benchmarking our human endothelial angiogenic model to the gold standard laser-induced choroidal neovascularization (CNV) model, we studied Rohlenova et al.’s transcriptome profiling of endothelial cells in murine CNV [73]. We recognized that our wet AMD BOECs subpopulations might not converge exactly to the distinct phenotypes of murine CNV in vivo. However, there were some resemblances of our human BOEC cell states to the in vivo tip endothelial cells, proliferating endothelial cells and angiogenic endothelial cells of post-capillary venule origin, etc. We curated a few markers from the CNV tip endothelial cells, namely *TGIF1*, *PGF* and *RHOC*. When we visualized these genes in our BOEC dataset, activated BOECs appeared to express relatively higher levels of *TGIF1* and *PGF* than the other populations, even though we did not resolve major differences between wet AMD and normal BOECs (Figure 3a). *TGIF1* (TG-interacting factor 1) is a regulator of stress-induced proinflammatory phenotype in human vascular endothelial cells [74], which was in consensus with the proinflammatory tumor necrosis factor signaling in activated BOECs, potentially inducing the specialization of tip cell-like phenotypes in this population.

Collagen biosynthesis was an upregulated pathway in angiogenic endothelial cells of CNV [73]. We analyzed collagen genes, *COL4A1* and *COL18A1*, and matrix metalloproteinases, *MMP2* and *MMP14* (Figure 3b). The expressions of these targets were generally higher in the primary populations compared to the evolved populations, supporting their involvement in vascular remodeling and cell proliferation. In particular, the expression levels of *COL18A1* and *MMP14* were elevated in wet AMD BOECs compared to their normal counterparts (Figure 3b). COL18A1 (C-terminal fragment being endostatin) is a heparan sulphate proteoglycan localized to endothelial basement membrane [75,76]. Overexpression of *COL18A1* in mice leads to ultrastructural changes, disorganization and loosening of the basement membrane [77], while upregulation of *MMP14* contributes to a change in extracellular matrix composition/stiffness that promotes a more angiogenic phenotype of endothelial cells [78,79]. Taken together, *COL18A* or *MMP14* might confer abnormal changes to the extracellular matrix integrity in wet AMD BOECs.

Notably, another CNV-associated gene, *VWF*, was pronouncedly increased in wet AMD BOECs than normal BOECs (Figure 3c). VWF is a well-known endothelial activation marker and is produced in response to mediators such as cytokines, epinephrine or vasopressin [80]. Plasma levels of vWF is also increased in wet AMD patients compared to controls [81,82]. We hypothesized that our wet AMD BOECs might recapitulate some of the hallmarks of injury response in patients. Wet AMD BOECs also showed the same trend by having higher expression of *KDR*, the vascular endothelial growth factor receptor 2, than normal BOECs (Figure 3c). Trafficking of intracellular KDRs to the cell surface increases the sensitivity of endothelial cells to VEGFA [83]. Therefore, wet AMD BOECs might have increased sensitivity to angiogenic stimuli.

## 3. Discussion

We analyzed a single-cell transcriptomic dataset of wet AMD and normal control endothelial cells that had been induced to undergo angiogenic spouting. Our human sprouting BOEC model shares some common traits with the murine laser induced CNV model [73]. Firstly, we found that ‘activated’ BOECs showed proinflammatory hallmarks and were transcriptomically more like the tip-like cell population in CNV in vivo. CNV in wet AMD is known to be associated with an inflammatory milieu, involving cytokines and complement system activation [84,85]. Secondly, extracellular matrix regulation of endothelial phenotypes was apparent in our wet AMD BOECs and CNV in vivo. Perturbations to collagen biosynthesis and turnover could induce basement membrane changes that promote proangiogenic activity of endothelial cells. Finally, wet AMD BOECs possessed markers of increased sensitivity to angiogenic and other stress paradigms. Transcriptomic analyses of human tissue with cerebrovascular malformations, such as cerebral cavernous malformations and brain arteriovenous malformations, have similarly identified enriched pathways associated with angiogenesis, including endothelial cell migration and extracellular matrix remodeling [86,87,88].

We have also provided new insights that were previously not known to be implicated in CNV. Angiogenesis in our wet AMD BOECs was driven by interleukin signaling and amino acid metabolism. While these processes have not been well reported in the pathogenesis of CNV, our wet AMD BOECs reflected pathological angiogenesis mechanisms observed in other diseases with vasculopathy or aberrant vessel growth, such as in tumor angiogenesis [89,90,91]. On the other hand, the normal control BOECs featured physiological angiogenesis mediated by VEGF, PDGF and Eph-Ephrin signaling. The main disparity of our BOEC model from the mouse CNV model is that laser induced CNV stimulates acute angiogenic response [13,18], whereas our patient-derived BOECs can capture the human genetic complexity and retain some age-related disease hallmarks in vitro [31,32,92,93].

We acknowledge the limitations of this study, including a small sample size of donors and lack of functional validation. The derivation of human BOECs does not involve organotypic differentiation for the cells to behave exactly like choroidal endothelial cells. Nevertheless, our study has elucidated a few interesting endothelial processes that may underpin pathological angiogenesis in wet AMD, warranting further investigations. To this end, we have created an interactive web-based visualization of the sprouting BOEC dataset (https://christinecheunglab.shinyapps.io/human_wet_AMD_sprouting/, accessed and last updated on 10 October 2022), to provide a transcriptomic resource for discovery research.

## 4. Materials and Methods

### 4.1. Patient Selection and Sample Collection

Participants (2 wet AMD and 1 normal subjects) were enrolled from the retina clinic of the Singapore National Eye Center. Inclusion criteria were age 40–80 years (demographics detailed in Appendix A). Participants with wet AMD were specifically diagnosed to have polypoidal choroidal vasculopathy (PCV) based on the presence of polypoidal dilatations in indocyanine green angiography (ICGA). A healthy volunteer was selected for the control group based on the lack of AMD or PCV from clinical examination. To collect blood samples for downstream analysis, 10 mL of fresh blood specimen was obtained from each participant and processed in the laboratory within 6 h. Upon Ficoll^®^ Paque (GE Healthcare, Chicago, IL, USA) centrifugation of the blood sample, the resulting buffy coat layer which contained peripheral blood mononuclear cells (PBMCs) was isolated from which DNA extraction was performed for genotyping with the OmniExpress chip. The remaining PBMCs were cultured for the derivation of blood outgrowth endothelial cells (BOECs).

### 4.2. Derivation and Maintenance of Blood Outgrowth Endothelial Cells

BOECs were generated as previously described [94] with modifications. One volume of peripheral blood samples (5–9 mL per participant) was diluted with 1 volume of phosphate-buffered saline (PBS) and the buffy coat layer was separated via density gradient centrifugation with Ficoll^®^ Paque (GE Healthcare). The buffy coat, which was enriched with peripheral blood mononuclear cells (PBMCs), was carefully collected, washed with PBS, resuspended in heparin-free, EGM-2 medium (Lonza, Basel, Switzerland) supplemented with 16% defined fetal bovine serum (FBS; Hyclone, Logan, UT, USA) and counted. Plasma was also collected and stored at −80 °C. Subsequently, the PBMCs were seeded into collagen I-coated well(s) according to a cell density of ≥1.5 × 10^6^ cells/cm^2^. Medium was replaced every two to three days. Outgrowth endothelial colonies should emerge visually between seven to 14 days after seeding. The cells were expanded to the third passage before any applications were performed, including phenotyping and functional evaluation, to opt out unwanted leukocytes. After the third passage, BOECs were cultured on collagen-I-coated tissue culture dishes in heparin-free, EGM-2 with 10% heat-inactivated FBS and medium was replaced every two to three days.

### 4.3. Fibrin Gel Bead Sprouting Assay

To subject BOECs to a sprouting context, a fibrin gel bead sprouting assay was performed as per described [95] with modifications. BOECs were coated onto Cytodex 3 microcarrier beads (Merck, Germany) at 150 cells per bead with agitation for 4 h and allowed to adhere to the beads overnight. Coated beads were subsequently suspended in 2 mg/mL fibrinogen solution (Merck, Germany) at a concentration of 500 beads/mL and clotted with 0.625 U/mL thrombin (Merck, Germany). Heparin-free EGM-2 with 10% FBS (Gibco) was added to the clotted gels containing the cell-coated beads, and cells were incubated overnight at 37 °C and 5% CO_2_ for 24 h to allow sprout formation.

To process fibrin gel sprouted BOECs for single-cell RNA sequencing, gels were rinsed with 1×PBS once, dislodged with a small spatula, and agitated for 15 min at 37 °C/5% CO_2_ in 50 FU/mL of nattokinase in 1×PBS. The resulting cell/bead suspension was collected and filtered through a 37 μm mesh strainer before being washed with heparin-free EGM-2 with 10% FBS. Subsequently, beads were resuspended with trypsin and agitated for 5 min at 37 °C/5% CO_2_. The cell suspension was then filtered through a 37 μm mesh strainer again, pelleted, and resuspended in heparin-free EGM-2 with 10% FBS, before being counted for sequencing purposes.

To capture images of representative 24 h sprouting BOECs, fixing, permeabilization, staining and image acquisition of the gel-embedded BOECs were performed as described in [31]. Briefly, the gels containing sprouted BOECs were fixed with 4% paraformaldehyde, permeabilized in 0.5% Triton X-100, stained for actin filaments using TRITC-Phalloidin in 1×PBS containing 1% BSA, and stained for nuclei with 500 ng/mL DAPI. Gel-embedded BOECs were imaged using an inverted laser scanning confocal microscope (LSM800, Carl Zeiss) using a Plan-Apochromat 20x/0.80 objective lens. Two-channel Z-stack images (AF568 and DAPI) of whole beads were captured using the ZEN software (blue edition, Carl Zeiss), acquiring images of 1024 × 1024-pixel resolution from 0.6× optical zoom at Z-intervals of 1.11 µm. For each donor BOEC line, two to four of the most representative well-formed individual filopodia were captured.

### 4.4. Single-Cell RNA Sequencing and Analysis

Each donor’s BOECs were sprouted in five wells of fibrin gels, with approximately 200 beads per well. After 24 h of sprouting assay, the BOECs embedded in fibrin gels were dissociated, pooled and split into two cell suspensions for loading onto 10× Genomics Chromium Controller chip by facility personnel at Single-cell Omics Centre (SCOC), Genome Institute Singapore (GIS). Hence, each donor’s sprouting BOECs were prepared as two separate scRNA-seq libraries using Chromium Single Cell 3′ v3 Reagent Kit (10× Genomics) by SCOC GIS and the final ready-to-sequence libraries were handed over with quantification and quality assessment reports from Bioanalyzer Agilent 2100 using the High Sensitivity DNA chip (Agilent Genomics). Individual libraries were pooled equimolarly and sent for sequencing by NovogeneAIT Genomics (Singapore). Raw sequencing data was also processed by NovogeneAIT Genomics (Singapore) using CellRanger (10× Genomics) with reads mapped to the human genome assembly (GRCh38).

Using the output filtered matrix files, analysis of the single-cell RNA sequencing of the three BOEC samples was performed using the Seurat package (v 4.1.1) [96,97]. Low-quality cells were filtered by excluding cells with low number of detected genes (nGene; less than 200) and transcript counts (nUMI; less than 500) based on inspection of the spread of these parameters per sample. To exclude potential doublets/multiplets, cells with abnormally high gene number (more than 9500) were filtered out, and *DoubletFinder* (v 2.0.3) [98] (expected doublet rate of 5%, 1:10 PCs and pK of 0.06) was applied to predict and remove doublets. The *paramSweep_v3* function and mean-variance-normalized bimodality coefficient (BCmvn) were used to determine the optimal pK value for the dataset. Furthermore, we excluded cells with high percentage of mitochondrial genes (greater than 20%), a threshold based on a previously reported mitochondrial gene percentage in endothelial cells [99]. We also excluded cells with low complexity of gene expression (less than 0.775), measured by the ratio of number of genes to number of transcripts (log[nGene]/log[nUMI]), as well as cells with low percentage of ribosomal reads (less than 5%). Finally, gene-level filtering was performed to exclude genes expressed in less than three cells.

The filtered dataset was scaled and normalized using *SCTransform* on each sample, with the parameter of variables to regress set as mitochondrial gene percentage, before integration based on 3000 integration features. After running a principal component analysis (PCA) and *RunUMAP* at 1:30 dimensions, *FindNeighbours* at 1:30 dimensions and *FindClusters* at a resolution of 0.1 were used to cluster the cells in the integrated dataset. A clustering tree was also generated using the *clustree* (v 0.4.4) function [100], which depicts the relationships between clusters as the resolution increases.

### 4.5. Cluster Marker Identification and Differential Expression Analysis

To identify cluster marker genes, *FindConservedMarkers* was applied. Gene enrichment of the top 100 cluster markers sorted by log2 fold change was performed using the clusterProfiler package (v 4.2.2) [101] and ReactomePA package (v 1.38.0) [102] with the parameters pAdjustMethod = “BH”, pvalueCutoff = 0.05, and qvalueCutoff = 0.2. To perform differential expression analysis between wet AMD and normal BOECs for each cluster, *FindMarkers* using Wilcoxon Rank Sum test and log2FC threshold of 0.25 was used. The lists were further filtered to include *p.adj* values of less than 0.05. Gene enrichment of differentially expressed genes between wet AMD and normal per cluster was conducted using the clusterProfiler and ReactomePA packages.

## Figures and Tables

**Figure 1 ijms-23-12549-f001:**
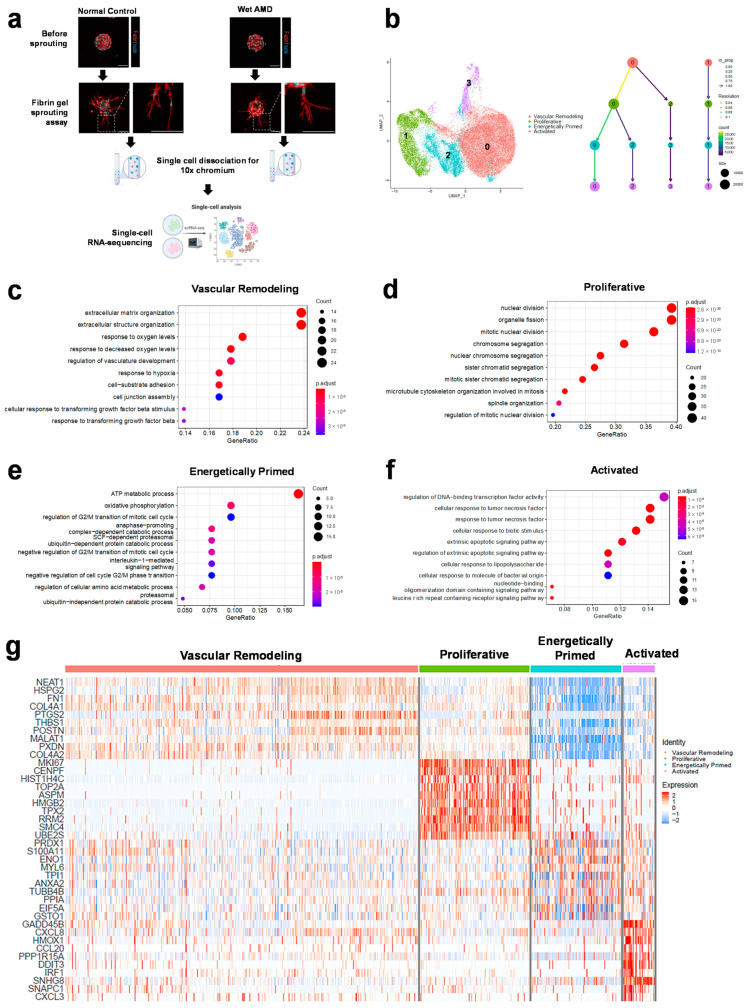
Human blood outgrowth endothelial cells adopt distinct cell states during sprouting angiogenesis. (**a**) Blood outgrowth endothelial cells (BOECs) derived from wet AMD and normal subjects were induced to sprout from cell-coated microcarrier beads in a fibrin gel sprouting angiogenesis assay, dissociated into single cell suspensions and subjected to single-cell RNA sequencing for downstream analysis. Images show confocal micrographs of representative normal and AMD BOECs at 24 h of the sprouting assay (red: f-actin; cyan: nuclei). Scale bar, 100 µm. (**b**) UMAP of integrated dataset showing derived endothelial clusters (**left**) and clustering tree depicting the relationships between clusters as resolution increases from 0.04 to 0.1 (**right**). Size of nodes (size) represents number of cells per cluster and color of nodes (Resolution) the resolution used to generate clusters. Numbers in each node represent cluster names in their corresponding resolutions. Edges are colored according to the number of cells from the incoming node (count). (**c**–**f**) Enrichment analysis of the top 100 cluster markers sorted by log2 fold change based on the Gene Ontology Biological Processes knowledgebase. Only top 10 enriched processes based on adjusted *p* values (p.adjust) are shown. Count indicates the number of genes within each term. (**g**) A heatmap of top 10 marker genes for each cluster and their corresponding expressions per cell.

**Figure 2 ijms-23-12549-f002:**
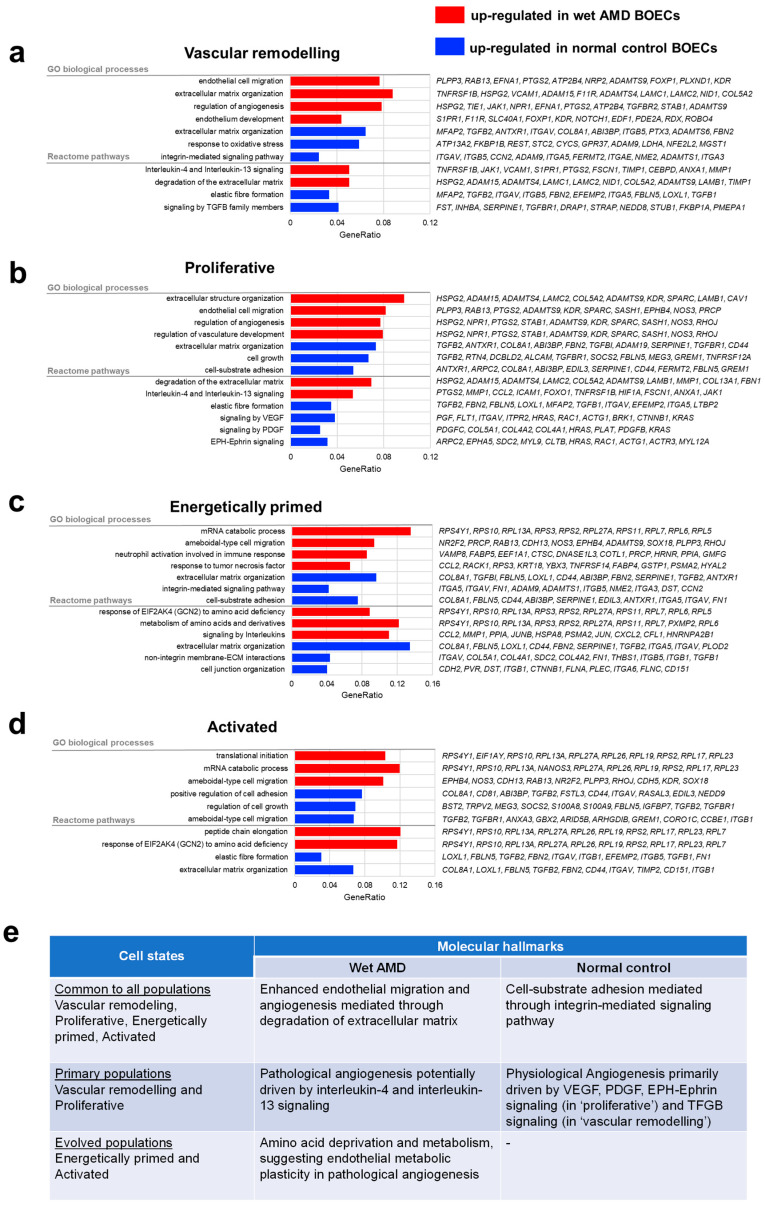
Wet AMD endothelial cells reveal pathological angiogenesis hallmarks. (**a**–**d**) Enrichment of top 100 differentially expressed genes (p. adj < 0.05) based on gene ontology biological processes and Reactome Pathway knowledgebases, for each of the four clusters as indicated. The top enriched processes and pathways are shown. Red bars represent wet AMD BOEC-upregulated processes, while blue bars represent normal BOEC-upregulated processes. (**e**) Summary of molecular hallmarks comparing wet AMD and normal BOECs that are common to all populations and found in primary/evolved populations.

**Figure 3 ijms-23-12549-f003:**
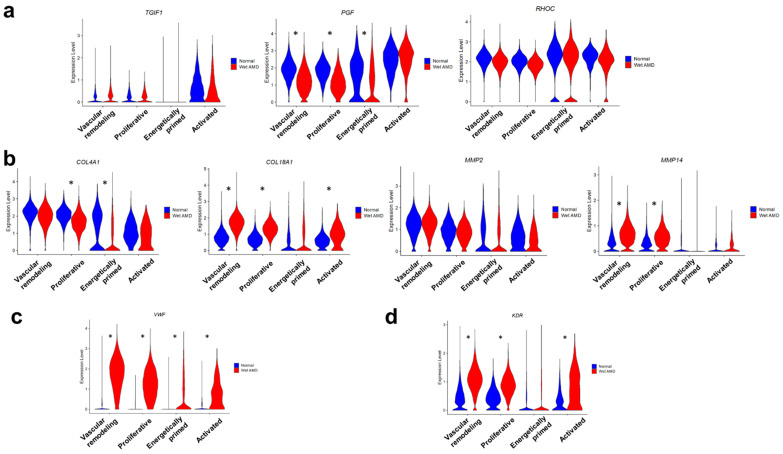
Benchmarking of human sprouting BOECs to in vivo choroidal neovascularization profiles. (**a**) Violin plots showing expressions of curated CNV tip endothelial cell markers based on Rohlenova et al. (**b**) Violin plots showing expressions of collagen genes and matrix metalloproteinases. (**c**) Violin plot showing expression of *VWF*, an injury marker, in normal and wet AMD sprouting BOECs. (**d**) Violin plot showing expression of *KDR*, a VEGF receptor, in normal and wet AMD sprouting BOECs. Asterisks mark significant differential expression between wet AMD and normal BOECs within each cluster, as determined using *FindMarkers*.

## Data Availability

The authors declare that all data supporting the findings of this study are available within the paper and Appendix A. Specifically, single-cell sequencing (scRNA-seq) dataset that support the findings of this study are available in the Gene Expression Omnibus repository, Accession number GSE213933, upon publication of this paper.

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
