# Peer review of "Single-Cell Transcriptome of Wet AMD Patient-Derived Endothelial Cells in Angiogenic Sprouting"

_ijms, 2022, doi:10.3390/ijms232012549_

Round 1

Reviewer 1 Report

Yeo and colleagues describe single-cell transcriptome of wet AMD patient-derived endothelial cells in angiogenic sprouting. They generated blood out-growth endothelial cells (BOECs) from wet AMD and normal control subjects and subjected the cells to a 3D angiogenesis assay. From these cells they performed single-cell sequencing and compared their data set to published in vivo mouse data sets.

The manuscript is well written, the data are clearly presented, the limitations of the study identified correctly.

I have only two minor comments:

Line 29: the provided link is not working

Figure 1a: the presented picture suggests that the wet AMD cells sprout less.

Reviewer 2 Report

Yeo et al. realized a very interesting article describing the “Single-cell transcriptome of wet AMD patient-derived endothelial cells in angiogenic sprouting”. I consider the manuscript very interesting but, at the same time, I suggest several revisions needed to improve the reliability and the completeness of the paper: 

·      The study suffers of a little sample size (2 cases and 1 control), and should be widened or, at least, underlined as a limit.

·      Are experiments realized at least in triplicate?

·      Chapter 4.4. The statistical analysis should be shifted to a separate chapter.

·      The “Discussion” section should be improved. For example, I suggest comparing produced data with results obtained from other NGS studies involving pathologies with an important vascular component, such as CCM and MAV. The recent PMID: 32877751, PMID: 32184807 and PMID: 30523548 could represent a substrate able to enforce the role of considered cellular mechanisms.

·      Finally, manuscript requires English revisions and typos correction. 

Round 2

Reviewer 2 Report

The authors addressed al suggested points.